# Bioguided Phytochemical Study of *Ipomoea cairica* Extracts with Larvicidal Activity against *Aedes aegypti*

**DOI:** 10.3390/molecules27041348

**Published:** 2022-02-16

**Authors:** Víctor Álvarez Valverde, Gerardo Rodríguez Rodríguez, Silvia Argüello Vargas

**Affiliations:** 1Programa Regional en Ciencias Veterinarias Tropicales, Escuela de Medicina Veterinaria, Universidad Nacional, Heredia 40101, Costa Rica; silvia.arguello.vargas@una.cr; 2Laboratorio de Fitoquímica, Escuela de Química, Universidad Nacional, Heredia 40101, Costa Rica; lafit@una.ac.cr

**Keywords:** chromatography, coumarins, secondary metabolites, organic extracts

## Abstract

Vector-borne diseases, such as those transmitted by *Aedes aegypti*, are a constant threat to inhabitants of tropical regions of the planet. Synthetic chemicals are commonly used as a strategy to control them; however, these products are known to persist in ecosystems and drive the appearance of resistance genes in arthropod vectors. Thus, the use of natural products has emerged as an environmentally friendly alternative in integrated vector control strategies. The present bioguided study investigated the larvicidal potential of *Ipomoea cairica* extracts, fractionated using thin-layer and open-column chromatography, because this species has been shown to exert larvicidal effects on the genus *Aedes*. The objective of this study was to evaluate the nonvolatile components in ethanolic extract of *I. cairica* stems as a potential natural larvicidal, and coumarins, such as 7-hydroxy-6-methoxychromen-2-one (scopoletin) and 7-hydroxychromen-2-one (umbelliferone), were identified as major compounds; however, they were not shown to be responsible for the larvicidal activity. Based on the results of the larvicidal action tests, these coumarins are not directly responsible for the larvicidal activity, but this activity might be attributed to a synergistic effect of all the compounds present in the most active secondary fraction, called F.DCM, which had an LC_50_ value of 30.608 mg/L. This type of study has yet not been conducted in the region; therefore, it is an important contribution to recognizing a natural and easy-to-cultivate source of vector control, such *I. cairica*.

## 1. Introduction

*Aedes aegypti* is one of the most important vectors for the transmission of viral diseases, including dengue, Zika, chikungunya, yellow fever, and Mayaro [1,2,3]. In Costa Rica, 9438 cases of dengue were diagnosed (at epidemiological week 44) in 2020, and since 2010, more than 185,471 cumulative cases have been reported [4]. The chemical control of arthropod vectors has been used as an emergency strategy in the event of outbreaks; however, synthetic products, such as organophosphates, pyrethroids, and carbamates, are nonspecific and exhibit poor water solubility—they persist in ecosystems, and their inadequate use has caused the appearance of resistance in vectors in different parts of the world. Therefore, the use of natural larvicides has emerged as an alternative for the control of arthropod vectors, since natural secondary metabolites with multiple bioactivities, including larvicidal activities, have been identified and are reincorporated into ecosystems once their function is fulfilled [5,6,7,8,9,10,11]. Eco-friendly and effective mosquitocidal extracts have the cumulative advantages of being cost effective, environmentally benign, safe to nontarget organisms, and biodegradable [10].

Numerous studies have shown the contributions of essential oils and volatile fractions of plants as mosquito larvicides or repellents using specific extraction methods for this type of metabolite [10,12,13,14]. Nevertheless, few studies have focused on the nonvolatile fractions of plants. Since all the compounds have different functions in plants, an interesting approach is to study the nonvolatile fractions of plants with the aim of identifying some important bioactivities, such as larvicidal activity. Nevertheless, the scientific community must acknowledge the importance of research into bioinsecticides and their roles preventing the appearance of resistance.

*Ipomea cairica*, commonly known as morning glory, is an extremely common plant in tropical climates that is known for its rapid growth and abundant spread, and it is mainly used for ornamental purposes. It contains diverse secondary metabolites with reported bioactivities, including larvicidal, anticancer, antimicrobial, antinociceptive, and cytotoxic activities [5,6,7,9,15,16,17]. A preliminary study determined that, of a set of 14 plants found in Costa Rica, 5 showed larvicidal activity against *A. aegypti*, with the ethanolic extract of *I. cairica* stems representing the most lethal extract toward *A. aegypti*, with a 50% lethal concentration (LC_50_) of 0.0341 mg/mL and a 95% confidence interval of 0.0293–0.0393 mg/mL [13]. The objective of the current project was to evaluate the nonvolatile components in the ethanolic extract of *I. cairica* stems as potential natural larvicidal agents.

## 2. Results

Table 1 shows the larvicidal activity of the ethanolic extracts of stems and leaves. The ethanolic extract of *I. cairica* stems produced 71.3% mortality, and that of leaves produced 40.0% mortality. This observation is consistent with the findings of a preliminary study [15]. Since the ethanolic extract of leaves was not as promising as that of stems in terms of biological activity, the present study focused solely on *I. cairica* stems.

The larvicidal activity of the ethanolic stem extract and its increasingly polar fractions obtained using column chromatography (CC) are shown in Table 2. When applied at 100 mg/L, the crude ethanolic extract of *I. cairica* stems presented a relatively high larvicidal activity of 71.3% mortality. The acetone fraction (F.Ac) showed a low percentage of mortality, and hexane (F.He) and methanol (F.Me) fractions showed no larvicidal activity. In contrast, the dichloromethane fraction (F.DCM) showed 100% mortality when tested under similar conditions.

Figure 1 shows the thin-layer chromatography (TLC) analysis of the samples reported in Table 2. Each column numbered from 1 to 5 corresponds to a different sample, which moves from bottom to top with the mobile phase, allowing the separation of the compounds. Sample 1 is the original extract, and fractions 2–5 correspond to secondary fractions obtained with different solvents from a chromatographic column of the original ethanolic extract. TLC facilitated the separation of most of the compounds present in each evaluated sample. The TLC plate was visualized with ultraviolet light at a wavelength of 365 nm, and the absorption at these wavelengths indicates the presence of aromatic rings and/or conjugated double bonds in the structure of the metabolite. The band marked with the red arrow in Figure 1A indicates the presence of a metabolite with functional groups similar to those mentioned above. As the band intensity increased (Figure 1B) by spraying the plate with a base, such as KOH, the metabolite is highly probable to be a coumarin. Based on this information, we predicted that the secondary metabolites present in the samples run on this plate are very similar in terms of their chemical structure, and that at least two are coumarins.

The major compounds in the F.DCM were separated using preparative TLC, and each secondary fraction of the F.DCM was evaluated for larvicidal activity. All bioassays conducted with 100 ppm of the F.DCM showed larvicidal activity with 100% mortality; however, the subsequent fractionation of the F.DCM caused an absolute loss of larvicidal activity (Table 3).

Table 4 shows the results of the larvicidal activity of the F.DCM and that of the crude ethanol extract of the stems of *I. cairica*. The mean LC_50_ values were lower for the F.DCM than for the ethanolic extract of stems.

Figure 2 shows the larvicidal activity of the F.DCM and crude ethanolic extract (CE-Et) at different concentrations. At a 95% CI in each curve, the mortality response differed between the F.DCM and EC-Et. For a particular concentration ranging from 0 to 200 ppm, the observed percent mortality induced by the F.DCM was higher than that of EC-Et.

In Figure 3, two blue bands were observed for the F.DCM when the plate was visualized under UV light at 365 nm, which were intensified when the plate was sprayed with 10% KOH in ethanol. This result indicates the presence of coumarins in the plant stem.

Figure 4 shows the high-performance liquid chromatography (HPLC) chromatograms, recorded at 2 wavelengths: 280 nm, typical for molecules with aromatic rings, and 343 nm, which exclusively detects a few secondary metabolites, including coumarins. Each peak in the chromatogram represents a compound, indicating that the F.DCM consists of relatively few compounds, three of which are potentially coumarins.

As part of the process used to identify at least 2 secondary metabolites present in the active secondary DCM fraction, the signals or peaks present at 11.8 and 14.9 min in the chromatogram shown in Figure 4 were isolated, corresponding to the major peaks that absorb at 343 nm. The result of the isolation of the major peaks presented in the F.DCM that absorbs at 343 nm is shown in pure form in Figure 5 and Figure 6.

The HPLC chromatograms in Figure 7 show the agreement of the retention times between the standards of the 2 coumarins (B and C) and the F.DCM (A) under the same chromatographic conditions and at wavelengths of 280 and 343 nm.

Figure 8 and Figure 9 show the mass spectra of the two coumarins obtained from *I. cairica* stems. The mass spectra of the molecular ion were generated using this analysis in positive mode, i.e., the molecular weight of the +1 molecule, which in some cases may generate a series of additional peaks corresponding to various molecular fragments, as shown in Figure 8. This same figure shows the molecular ion at *m*/*z* 163, which corresponds to the molecular weight of the coumarin umbelliferone +1. Figure 9 shows a molecular ion at *m*/*z* 193, which corresponds to the molecular weight of the coumarin scopoletin +1.

The evaluation of the larvicidal activity of the coumarins was negative, suggesting that this activity might be related to the synergistic activities of all the compounds present in the F.DCM (Table 5).

## 3. Discussion

A preliminary study of larval mortality allowed us to establish the different larvicidal activities of ethanolic extracts of *I. cairica* stems and leaves [15], as shown in this study, where the ethanolic stem extract of this plant produced 71.3% mortality and the leaf extract produced 40.0% mortality. The photosynthetic tissues of plants are more concentrated in the leaves than in stems, and these tissues are extracted when in contact with organic solvents, such as ethanol. The resulting extract is rich in secondary metabolites and photosynthetic pigments, such as chlorophylls and xanthophylls, which not only hinder the separation of a particular secondary metabolite but also dilute the possible biological activity. In the present study, the ethanolic leaf extract was not as promising as the stem extract in terms of biological activity.

The crude ethanolic extract of *I. cairica* stems presents a relatively high larvicidal activity at 100 mg/L, with 71.3% mortality, compared with the larvicidal mortality data reported by Srivastava and Shukla (2015), who indicated that an essential oil extract of this plant achieves up to 100% mortality in 24 h when evaluated at 120 mg/L. Additionally, Chariandy et al. (1999) obtained positive mortality results 2 days after initiating the test by exposing stage IV *A. aegypti* larvae to 500 mg/L of an ethyl acetate extract of *Justicia pectoralis* for 1 h, ensuring that larvicidal activity was present in the initial extract before fractionation. The F.He and F.Me showed no larvicidal activity. In contrast, the F.DCM showed a mortality rate of 100% of the individuals when tested under conditions similar to the other samples. The low percent mortality of the F.Ac indicates that very few of the active secondary metabolites were eluted with this solvent, while none of these compounds were eluted by Me and He.

According to Ahbirami et al. (2014), *I. cairica* leaves have larvicidal activity against *A. albopictus* and *A. aegypti* when extracted with acetone or methanol, showing LC_50_ values of 101.94 mg/L and 105.50 mg/L, respectively, for *A. aegypti* [8]. The study by Ahbirami et al. is fundamental because it provides evidence that *I. cairica* possesses potential larvicidal activity; however, this study did not delve into different phytochemical approaches to determine whether acetone or methanol is the most appropriate solvent to extract the secondary metabolites responsible for larvicidal activity. An important difference between the extraction process used by Ahbirami et al. (2014) and the one performed in the present study must be noted. The difference lies in the extraction temperature, which several studies report to be a factor that must be meticulously regulated to improve the chemical integrity of the extract when temperatures no higher than 40 °C are used. Ahbirami et al. used temperatures up to 60 °C for at least 3 h; whereas, in the present study, the necessary precautions were taken not to exceed 40 °C, thus generating a possible fundamental difference in the extract composition [15,16].

Based on the results, the F.DCM comprises a combination of secondary metabolites that would be present in the crude ethanolic extract. The ethanolic extract contains a greater number of metabolites, some of which do not possess larvicidal activity. As a result, bioactive metabolites are present at a lower concentration in the EC-Et, which results in a lower bioactivity compared with the F.DCM.

The F.DCM showed 2 blue bands when visualizing the chromatographic plate under UV light at 365 nm, which were intensified when the plate was sprayed with 10% KOH in ethanol. This result indicates the presence of coumarins in the stem of the plant. Meira et al. (2012) [18] described the presence of two coumarins, 7-hydroxy-6-methoxychromen-2-one (scopoletin) and 7-hydroxychromen-2-one (umbelliferone), in the dichloromethane extract of *I. cairica.* These coumarins are identified by their retention factors (rf) of 0.25 and 0.45, respectively [19], which are within the range obtained in this study.

The similar retention times of the peaks of the standards and the peaks present in the HPLC chromatogram of the F.DCM suggests that the coumarins are scopoletin and umbelliferone. HPLC–mass spectrometry (MS) was used to verify this identification and showed that the molecular weights of the coumarins were consistent with those reported in the literature: 162.14 g/mol for umbelliferone [20] and 192.16 g/mol for scopoletin [21].

The evaluation of the larvicidal activity revealed no activity of these coumarins, suggesting that the larvicidal activity might be related to the synergistic effects of the components of the F.DCM. This larvicidal activity of the F.DCM was detected at relatively low concentrations compared with other studies [22,23]. An in silica study by Rollinger et al. (2004), analyzing approximately 110,000 molecules, suggests that analogs of scopoletin and its glycosylated homolog, scopoline, are notable as possible inhibitors of acetyl cholinesterase (ACh) [24], which is the same mechanism of action as organophosphates, such as temephos, used as larvicides in vector control [25]. Another recurring mechanism of action of nonpolar molecules is the suppression of adenosine triphosphate (ATP), which may be due to partial competition with nicotinamide adenine dinucleotide (NADH)-ubiquinone oxidoreductase [26,27]. A potential explanation for this finding is that F.DCM is enriched in components with a partially nonpolar profile that are present in the crude extract, and the structures of some of these molecules might have stereochemistry surrounding cycles that contribute to larval death [27]. Importantly, molecules that exhibit bioactivity must have target site specificity, which, in some cases, might be potentiated in the presence of synergistic components in the matrix [28]. Regarding natural larvicidal products, synergistic effects are commonly reported for volatile plant extracts [29,30,31] and sometimes for nonvolatile extracts [26,32]. In addition, plant molecules usually present low toxicity to mammals, and they are nontoxic to humans and other nontarget organisms at the recommended doses [33].

Because of both the thoroughness of the purification process and the larvicidal tests performed here, the larvicidal activity of the extracts and fractions obtained from the *I. cairica* stems were confirmed to have a comparatively high lethality with respect to other nonvolatile extracts of natural products evaluated against *A. aegypti* larvae [6,16,34]. Additionally, the larvicidal activity was not attributed to the pure compounds isolated from the active fractions; rather, this bioactivity was attributed to a synergistic effect of the set of compounds present in the ethanolic extract and those purified from the ethanolic extract present in F.DCM. Therefore, further studies are needed to corroborate whether the observed activity is linked to synergism. Additionally, this type of study has not yet been conducted in the region; therefore, it is an important contribution to recognizing a natural and easy-to-cultivate source of vector control—*I. cairica*.

## 4. Materials and Methods

The methodology described in the present study is based on a bioguided design, where the processes used to separate and purify components are based on the result of the larvicidal activity observed for each extract evaluated.

### 4.1. Plant Material

The sample was collected in San Miguel de Santo Domingo de Heredia, Costa Rica (permit no. R-012-2020-OT-CONAGEBIO). The leaves were separated from the stems, and both plant parts were dried in a convection oven at 40 °C for 72 h. Once dry, they were ground in a knife mill using a 1 mm sieve.

### 4.2. Sample Processing

A measure of 10 g of dried and ground *I. cairica* leaves and stems were exhaustively extracted with ethanol (Et) using an ultrasonic bath at 40 °C for 15 min for each extraction, and then the extracts were combined and dried using a rotary evaporator with a temperature-controlled bath at 40 °C under reduced pressure. Subsequently, 100 g of the stems were extracted and concentrated; once dry, the extract was fractionated using CC with silica gel as the stationary phase and an increasingly polar gradient of He, DCM, Ac, or Me as the mobile phase. The resulting fractions were concentrated to dryness.

### 4.3. Monitoring Using TLC and Isolation Using Preparative TLC

The resulting fractions were evaluated using TLC on silica gel with a fluorescent indicator as the stationary phase, and toluene:ether (1:1) saturated with 10% acetic acid as the mobile phase [17]. Ultraviolet light was used at 254 nm and 365 nm to develop the plates, possible coumarins were revealed by spraying the plates with 10% potassium hydroxide in ethanol [17].

The preparative TLC technique was performed under the same conditions (stationary phase and mobile phase) as the TLC procedure; however, it used a greater amount of sample because it was designed to isolate the compounds. The bands of interest were scraped, extracted with DCM, and subsequently dried to recover the isolated compounds. The extraction, fractionation, and larvicidal activity assays were performed in quadruplicate to exclude any false positives or negatives throughout the test.

### 4.4. High-Efficiency Liquid Chromatography with a Diode Array Detector and Mass Spectrometry (HPLC-DAD-MS/MS)

For the HPLC-DAD technique, a Shimadzu system, consisting of a DGU20A5 degasser, LC20AT pump, SPDM20A diode array detector, and a 50 µL manual loop injector, was used. The gradient used was 0.1% trifluoroacetic acid as mobile phase A and methanol as mobile phase B; the run began with 15% mobile phase A and ended with 100% mobile phase B in 20 min, with a flow rate of 1 mL/min, as shown in Table 6. The column used was a Phenomenex Luna C18 column (250 × 4.6 mm, 5 mm particle size, 100 Å pore size). For the semipreparative HPLC technique, an Agilent Pursuit column (250 × 10 mm, 10 mm particle size) was used at a flow rate faster than the analytical flow rate of 4.5 mL/min. The mass detector was used with the following operating conditions: flow rate—0.5 mL/min; polarity—electrospray ionization (ESI)+; carrier gas—30 psi; nebulizer—45 psi; drying gas—55 psi; capillary voltage—5.5 kV; collision gas—10 psi.

### 4.5. Larval Culture

Eggs of the *A. aegypti* Rockefeller strain, that is susceptible to insecticides, were stored on filter paper in a dry environment out of direct light under controlled environmental conditions. When larvae were needed, eggs were placed in pans with a sufficient amount of water to assure proper hatching. Conditions were standardized in every batch to avoid larval overcrowding. Proper feeding occurred to ensure healthy and homogeneous larval growth.

### 4.6. Bioassay of Larvicidal Activity

The method used to determine larvicidal activity was described by the World Health Organization in the WHO/VBC/81.807 report for the determination of larvicidal activity [25]. Third- and fourth-instar larvae of the *A. aegypti* Rockefeller strain, susceptible to insecticides, were used. The larvae were maintained in the laboratory without any exposure to known insecticides. The larvae were obtained from eggs hatched simultaneously and raised under the same food and environmental conditions to ensure physiological homogeneity in each assay. In the assays, 4 replicates were used per concentration of metabolite or extract, with 20 larvae each and a total volume of 25 mL. A positive control group treated with temephos and a negative control group treated with water were established. Mortality was determined 24 h after the start of the assay. The initially evaluated concentrations were 100 ppm for the determination of the LC_50_ and LC_90_ values. The CIs were determined for each value.

## Figures and Tables

**Figure 1 molecules-27-01348-f001:**
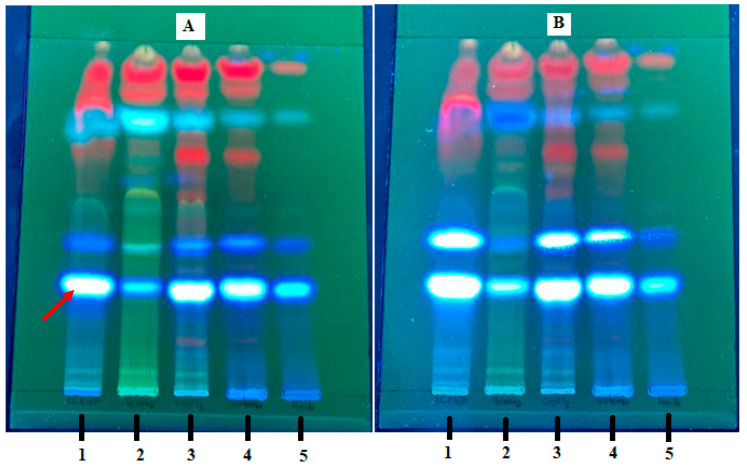
Thin-layer chromatography of the fractions listed in Table 3 that were evaluated for larvicidal activity visualized with UV light at 365 nm (**A**) and with KOH in ethanol (**B**); 1—crude extract in Et; 2—F.He; 3—F.DCM; 4—F.Ac; 5—F.Me.

**Figure 2 molecules-27-01348-f002:**
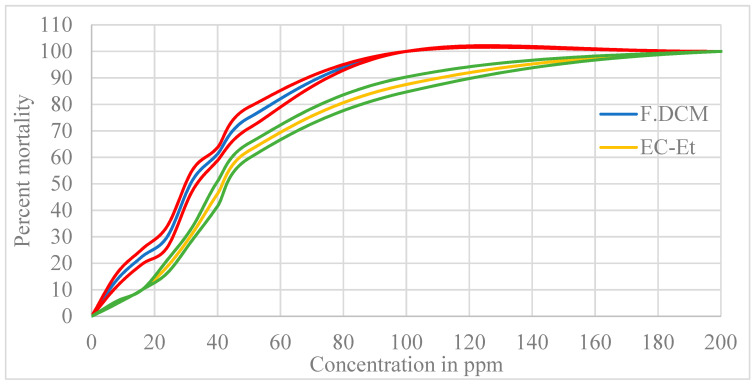
Observed percent mortality according to the sample concentration for the F.DCM and CE-Et shown in blue and yellow, respectively; the red and green lines indicate the expected mortality at the 95% CI.

**Figure 3 molecules-27-01348-f003:**
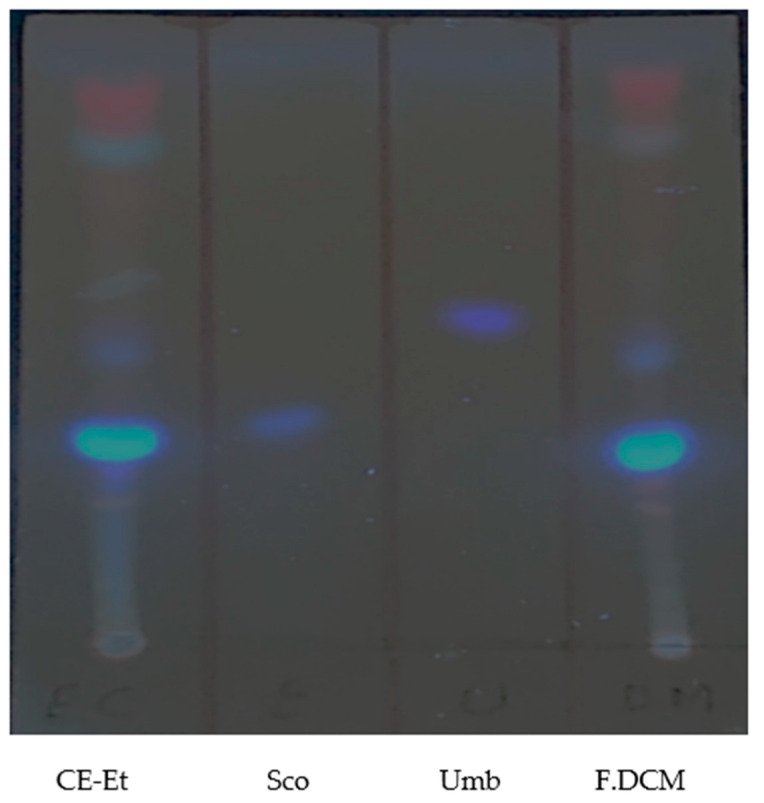
TLC plate eluted with toluene:ether (1:1) saturated with 10% acetic acid and developed with 5% KOH in ethanol; CE-Et—crude extract in ethanol; Sco—scopoletin standard; Umb—umbelliferone standard; F.DCM—dichloromethane secondary fraction.

**Figure 4 molecules-27-01348-f004:**
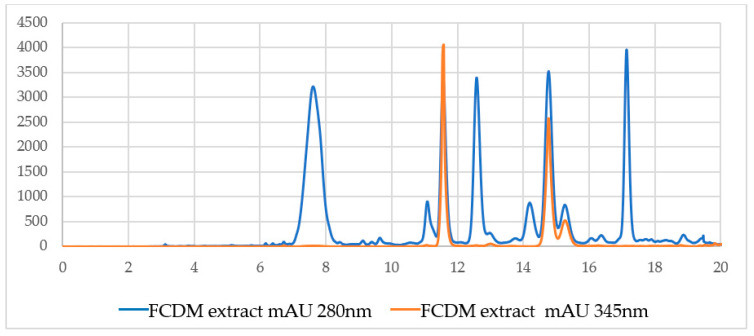
Reverse-phase HPLC diode array detector (DAD) chromatograms of the F.DCM obtained at 2 wavelengths, 280 nm and 343 nm.

**Figure 5 molecules-27-01348-f005:**
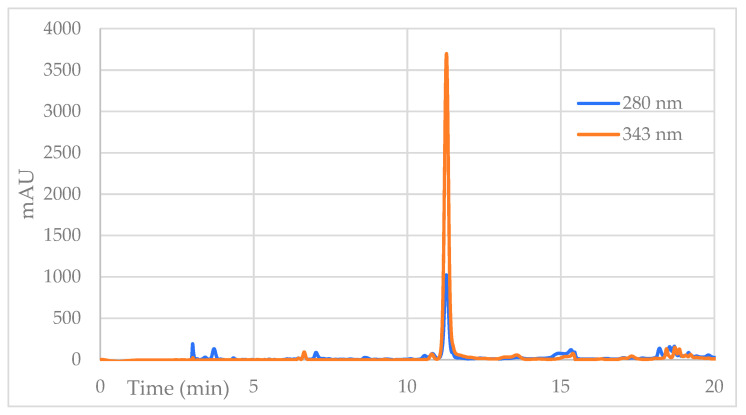
Reverse-phase HPLC-DAD chromatogram of the first purified compound at 2 wavelengths, 280 nm and 343 nm.

**Figure 6 molecules-27-01348-f006:**
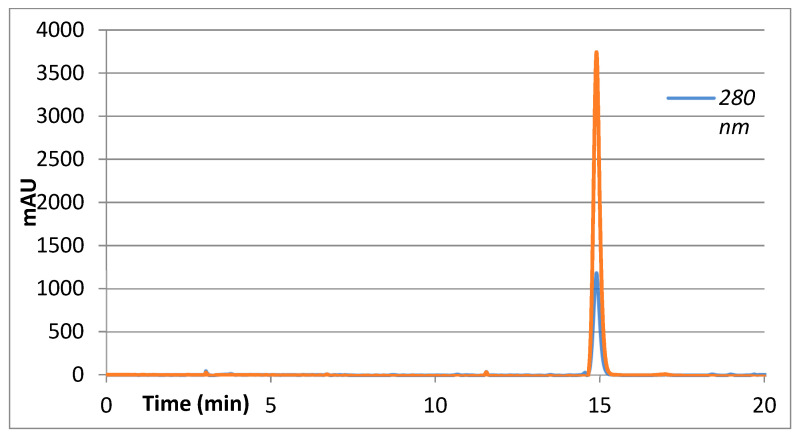
Reverse-phase HPLC-DAD chromatogram of the second purified compound at 2 wavelengths, 280 nm and 343 nm.

**Figure 7 molecules-27-01348-f007:**
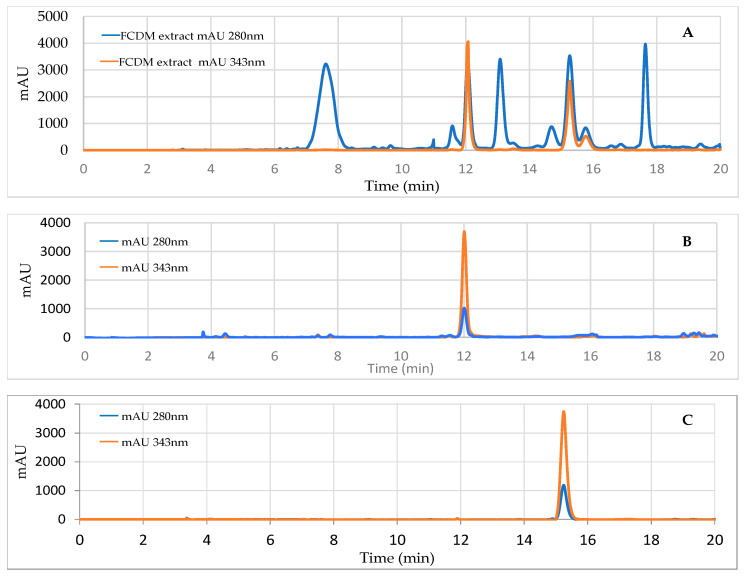
Reverse-phase HPLC-DAD chromatogram of the active F.DCM (**A**), scopoletin standard (**B**), and umbelliferone standard (**C**), at 2 wavelengths, 280 and 343 nm.

**Figure 8 molecules-27-01348-f008:**
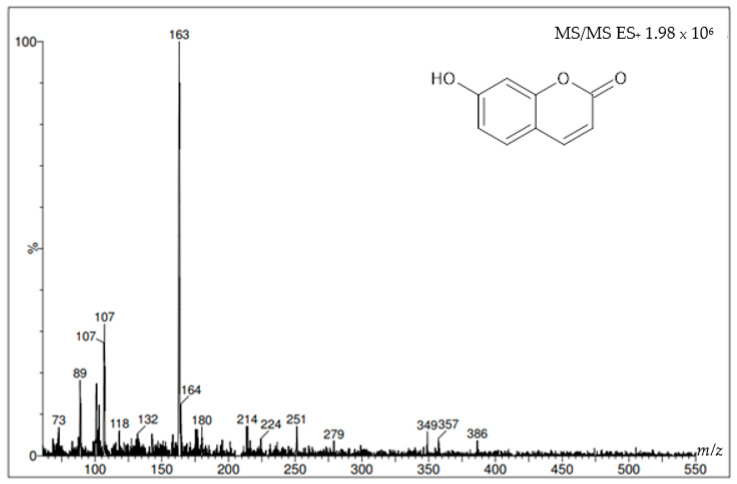
Mass spectrum with the molecular ion of the coumarin umbelliferone using HPLC-MS/MS spectrometer.

**Figure 9 molecules-27-01348-f009:**
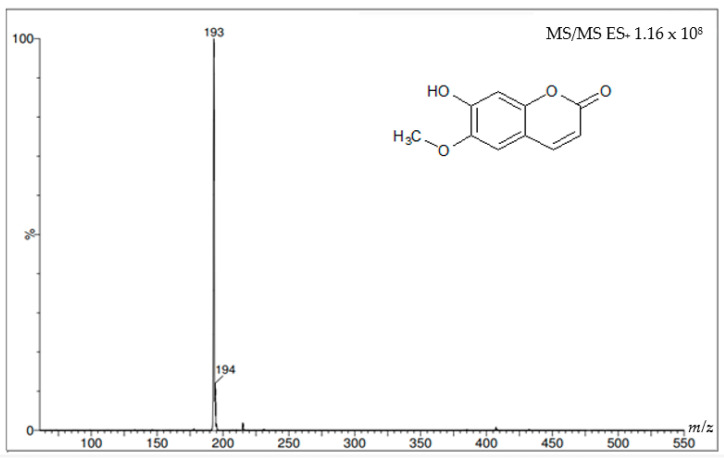
Mass spectrum with the molecular ion of the coumarin scopoletin HPLC-MS/MS spectrometer.

**Table 1 molecules-27-01348-t001:** Larvicidal activity of the ethanolic extracts of *I. cairica* stems and leaves.

Sample	Percent Mortality at 24 h
Stems	71.3 ± 4.8%
Leaves	40.0 ± 0.0%
Positive control	100 ± 0%
Negative control	0 ± 0%

**Table 2 molecules-27-01348-t002:** Larvicidal activity of the ethanolic extract and its increasingly polar fractions obtained using CC.

Sample	Percent Mortality at 24 h
Crude ethanolic extract	71.3 ± 4.8%
F.He	0 ± 0%
F.DCM	100 ± 0%
F.Ac	18.8 ± 2.9%
F.Me	0 ± 0%
Positive control	100 ± 0%
Negative control	0 ± 0%

F—fraction; He—hexane; DCM—dichloromethane; Ac—acetone; Me—methanol.

**Table 3 molecules-27-01348-t003:** Larvicidal activity of the F.DCM and its secondary fractions (1, 2, and 3) obtained using preparative TLC.

Sample	Percent Mortality at 24 h *
F.DCM	100 ± 0%
F.DCM (1)	0 ± 0%
F.DCM (2)	0 ± 0%
F.DCM (3)	0 ± 0%
Positive control	100 ± 0%
Negative control	0 ± 0%

* Mean values of four replicates.

**Table 4 molecules-27-01348-t004:** Results of the larvicidal activity of the F.DCM and crude ethanolic extract.

Fraction	LC_50_ in mg/L (95% CI *)	LC_90_ in mg/L (95% CI *)	Diagnostic Dose in mg/L (95% CI *)
F.DCM	30.608 (23.9–39.1)	79.875 (62.5–98.1)	349.2 (273.1–446.4)
Crude ethanolic extract	42.1 (32.1–55.0)	131 (100.0–172.0)	664 (507–870)

* CI—confidence interval; LC_90_—90% lethal concentration.

**Table 5 molecules-27-01348-t005:** Larvicidal activity of the isolated pure compounds scopoletin and umbelliferone, and of a combination of both compounds, isolated from the plant *I. cairica*. The test was performed with 4 homogeneous replicates of *n* = 80 larvae.

Sample	Percent Mortality
Scopoletin	0
Umbelliferone	0
Scopoletin + Umbelliferone	0
CONTROL+	100.00
CONTROL−	0.00

**Table 6 molecules-27-01348-t006:** Gradient used in reverse-phase HPLC for the chromatographic separation of the secondary metabolites present in the crude extracts of *I. cairica*.

Time (min)	Concentration of A	Concentration of B	Flow Rate (mL/min)
0.01	85	15	1.00
3.00	65	35	1.00
7.00	60	40	1.00
10.00	55	45	1.00
13.00	50	50	1.00
15.00	50	50	0.50
16.00	50	50	0.50
18.00	15	85	0.50
18.00	15	85	1.25
19.00	0	100	1.25
20.00	85	15	1.00

## Data Availability

Data supporting reported results are available at victor.alvarez.valverde@una.cr.

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
