# Peer review of "Bioguided Phytochemical Study of Ipomoea cairica Extracts with Larvicidal Activity against Aedes aegypti"

_molecules, 2022, doi:10.3390/molecules27041348_

Round 1
Reviewer 1 Report
The article although interesting and the methods are sufficiently implemented,
it lacks novelty and important findings. The research resembles more a short communication, as the authors state their preliminary results but fail to communicate the impact and understand that more work has to be done.
Author Response
In the present document you will be able to find important improvements regarding to deficiencies that were pointed out in the original submited document. It becomes neccesary for me to express that there are two important idieas in the document that makes it publishable: 1) the relevance of working with non-volatile extracts, this is due to the fact that most studies on larvicidal activity focus on essentail oils and volatile fractions, this is not the case and 2) this kind of studies have not been cunducted in this region, therefor it is an important contibution to the region to recognize a natural and easy to cultivate source of vector control.

Reviewer 2 Report
Reviewer comments
Manuscript title: Bioguided phytochemical study of Ipomoea cairica extracts with larvicidal activity against Aedes aegypti
The manuscript entitled ‘Effect of phytochemical study of Ipomoea cairic discusses the prospective application of phytochemicals against the dengue vector. However, there are some questions and corrections to be carried out to make the manuscript for further consideration
In brief: A complete check for formatting and typographic errors must be done. I can’t understand several points because of language barrier.
Specific points:
- Lines 12-14 in abstract needs to be modified to deliver a proper meaning. i.e. Synthetic chemicals are commonly used as a strategy to control them; however, these products are known to persist in ecosystems and can cause selection of resistance genes in arthropod vectors. -what does it mean?
- An improved set of keywords shall be chosen – the ones not in the manuscript title.
- A clear picture of mosquito resistance and advantage of bio-insecticide should be provided. At least refer following documents (10.3389/fphys.2019.01591, 10.1016/j.envint.2017.12.038, 10.3390/molecules26123695)
- I’m not convinced with method. Please prove detailed methodology for each experiment. Where is the mosquito culture? What about bioassay?
- Discussion shall be rewritten focusing only on the significant results and not all the tests conducted and random assay. The discussion section is very concise and does not include the possible mechanisms of the effects of PHYTOCHECMIALS.
- A proper explanation of obtained results with appropriate citations indicating all the results shall be included in the discussion
- Complete update of citations throughout from introduction to discussion is needed.
Author Response
In the present document you will be able to find important improvements regarding to deficiencies that were pointed out in the original submited document. It becomes neccesary for me to express that there are two important idieas in the document that makes it publishable: 1) the relevance of working with non-volatile extracts, this is due to the fact that most studies on larvicidal activity focus on essentail oils and volatile fractions, this is not the case and 2) this kind of studies have not been cunducted in this region, therefor it is an important contibution to the region to recognize a natural and easy to cultivate source of vector control.
About specific points:
1. improved in document
2. improved in document
3. improved in document
4 and 5. improved in document: now the document has a section where you can find the metodology used in each section, nevetheless it has to be consider that this is an bioguieded study, therefor it has to have as much detail as possible in order to fully understand the study.
6. Proper explanation included in document
7. Citations were improved in document
hope the improvements in the document meet your expectations!

Reviewer 3 Report
The paper of Valverde deals with the larvicidal effects on the genus Aedes. The manuscript reports ethanolic extraction followed by methanolic, hexane, acetone, and dichloromethane fractionation of I. cairica. As a second aim of their study, author reported that, larvicidal activities of I. cairica extract and fractions revealed that the crude ethanolic extract of I. cairica stems at 100 mg/L presented a relatively high larvicidal activity of 71.3% mortality. The acetone fraction showed a low percent mortality, and the hexane and methanol fractions showed no larvicidal activity. In contrast, the dichloromethane fraction showed 100% mortality when tested under similar conditions.
However, I am much more concerned about major weakness that may strongly limit the acceptance of this study. The objective of the present study, as mentioned, was to identify the compound/s that produce larvicidal activity in the fractions of ethanolic extracts of I. cairica stems. However, the major compounds of the dichloromethane fractions, scopoletin and umbelliferone, was evaluated for larvicidal activity, and have shown no larvicidal activity at all, therefor the authors suggested that this activity could be related to the synergistic activity of all the compounds present in the dichloromethane fraction which is considered inconsistent with the objectives of the study.
Author Response
Hello dear Reviewer
I extend to you the humble and deepest apologise, there was a mistake in the selection of words that lead to the objective in the first verstion of the document submited. Therefor you will be able to find an improvement in the objective that shuould have been presented in the first document since the original document among with other specific corrections sugested by the other reviewers . Hope this change meets your expectations.

Round 2
Reviewer 1 Report
The authors have addressed the issues of the manuscript thus significantly improving it.
Though the introduction needs to be checked again especially for syntax errors, and the two ideas in the authors response that according to them make it publishable have to be incorporated clearly in the manuscript.Author Response
All spelling and syntax errors as well as other comments were properly addressed.

Reviewer 2 Report
The author revised the manuscript accordingly
Author Response
All spelling and syntax errors as well as other comments were properly addressed.

Reviewer 3 Report
The objective of the study has been reframed to evaluate the non-volatile components in ethanolic extract of I. cairica stems as a potential natural larvicidal through bioguided fractionation studies, which seems to be acceptable in the current form
Author Response

(The authors gave the same response as above.)
